# Separation of Mercury(II) from Industrial Wastewater through Polymer Inclusion Membranes with Calix[4]pyrrole Derivative

**DOI:** 10.3390/membranes12050492

**Published:** 2022-04-30

**Authors:** Iwona Zawierucha, Anna Nowik-Zajac, Jakub Lagiewka, Grzegorz Malina

**Affiliations:** 1Institute of Chemistry, Jan Dlugosz University in Czestochowa, 42-200 Czestochowa, Poland; a.zajac@ujd.edu.pl (A.N.-Z.); jakub.lagiewka@doktorant.ujd.edu.pl (J.L.); 2Department of Hydrogeology and Engineering Geology, AGH University of Science and Technology, Mickiewicza 30, 30-059 Cracow, Poland; gmalina@agh.edu.pl

**Keywords:** mercury, industrial wastewater, polymer inclusion membranes, removal efficiency

## Abstract

Polymer membranes with immobilized ligands are encouraging alternatives for the removal of toxic metal ions from aquatic waste streams, including industrial wastewater, in view of their high selectivity, stability, removal efficacy and low energy demands. In this study, polymer inclusion membranes (PIMs) based on cellulose triacetate, with a calix[4]pyrrole derivative as an ion carrier, were tested for their capability to dispose mercury (Hg(II)) ions from industrial wastewater. The impacts were assessed relative to carrier content, the quantity of plasticizer in the membrane, the hydrocholoric acid concentration in the source phase, and the character of the receiving phase on the performance of Hg(II) elimination. Optimally designed PIMs could be an interesting option for the industrial wastewater treatment due to the high removal efficiency of Hg(II) and great repeatability.

## 1. Introduction

Aquatic waste streams contaminated with mercury ions may generate a considerable environmental problem across the world due to the harmful effects of Hg species on human health and ecosystems [1]. The fast expansion of the industry globally has resulted in serious environmental problems of mercury in water; Hg(II) was numbered at sixth on the list of most toxic chemicals, and it is considered as one of the most hazardous and predominant heavy metals in Earth’s waters [2]. The major sources of Hg in the environment include plastics, oil, pulp and cement industries. Other sources may comprise used cells and fluorescent lamps containing Hg [3]. According to Polish legal requirements, the permissible concentration of mercury in groundwater and drinking water is of 0.001 mg/dm^3^ while 0.03 mg/dm^3^ is permissible in treated wastewater [4].

Numerous standard methods exist for the separation of mercury from wastewater, which include ion exchange, precipitation and adsorption [5,6,7]. The common drawback for most of them is associated with high exploitation and conservation costs, as well as the formation of hazardous effluents and sludge, the consumption of chemicals, the inability to reuse Hg and the difficulties in running multi-steps processes [8]. 

Membrane processes are an alternative to conventional methods of separating toxic metal ions from wastewater owing to their high efficiency with relatively small energy demand and the possibility of carrying out the process at ambient temperatures. The novel types of membranes, such as immobilized polymer membranes—Polymer Inclusion Membranes (PIMs)—show qualities worthy of special attention [9,10,11,12]. These membranes are obtained as a result of the physical immobilization of an ion carrier together with a plasticizer in a polymer matrix [13]. The role of selective ion carriers can be played by ionizable macrocyclic compounds, which have the ability to "recognize" ions and show specific complex-forming properties [9]. The features that characterize PIMs include considerable durability, chemical resistance, mechanical strength and operational stability, while the extractions of substances from the source phase, diffusion of the resulting complex through the membrane and re-extraction to the receiving phase are responsible for the transport of ions [14,15]. An essential property of PIMs is the ability of altering their composition, which can influence the efficiency and selectiveness of metal ions separation, taking into account the optimal process and environmental conditions [14]. *o*-NPOE was selected as the plasticizer, i.e., a solvent for an ion carrier (extractant), which increases flexibility and permeability of the membrane due to its low viscosity and a high dielectric constant (ε) of 23.1 [16]. The CTA polymer is, in turn, responsible for the mechanical strength of the membrane [14]. An exceptional benefit of CTA membranes is undoubtedly their hydrophilicity, which is accountable for the amount of volumetric flow and relatively low tendency to fouling [17]. The transport and separation properties of PIMs enable the removal of metal ions at the location of their generation [18], which is particularly beneficial from the point of view of minimizing the hazards of the aquatic environment.

To transport mercury ions through PIMs, various extractants were used: Cyanex 471X (tri-isobutylphosphine sulphide) [19], diphenylthiocarbazone (dithizone) [20,21] and 1-(2-pyridylazo)-2-naphthol (PAN) [21] as well as ionic liquids such as trioctylmethylammonium thiosalicylate (TOMATS) [22,23,24] and trioctylmethylammonium salicylate (TOMAS) [24].

Calixpyrroles can be potential extractants for use in PIMs thanks to ready-made methods of their synthesis, often high performance and relatively simple modifications of their complexing characteristics by functionalization by introducing relevant substituents [25,26]. Calixpyrroles represent macrocyclic compounds consisting of pyrrole elements coupled by quaternary carbon atoms at their 2,5-positions [27]. These ligands have gained considerable interest due to their capacity to bind anions, their ability to function as ditopic (ion-pair) receptors and their ability to host neutral molecules that adopt NH hydrogen bonds [28].

In the current study, we analyze the separation of Hg(II) ions by the PIM-containing a calix[4]pyrolle derivative as the ion carrier. To evaluate the performance of Hg(II) separation, we used the following parameters: the impact of the carrier concentration, the quantity of plasticizer in the membrane, the acidity of the source phase, the type of the strip phase and the feasibility of reusing the membrane.

## 2. Materials and Methods

### 2.1. Reagents

Inorganic reagents, i.e., mercury (HgCl_2_) and sodium (NaCl) chlorides, hydrochloric acid (HCl) and potassium iodide (KI) provided by POCh (Gliwice, Poland) and organic chemicals, i.e., cellulose triacetate (CTA), *o*-nitrophenyl octyl ether (*o*-NPOE) and dichloromethane (DCM) obtained from Fluka (Seelze, Germany), all of analytical grade, were used with no further cleansing. Double distillated water with a conductivity of 0.1 µS/cm was used to prepare aqueous solutions. 

### 2.2. Synthesis of Meso-octamethylcalix[4]pyrrole

The synthesis of *mes**o*-octamethylcalix[4]pyrrole (KP) (Figure 1) was described by Bayer [29] and Rothemund and Gage [30]. The structure of the synthesized *mes**o*-octamethylcalix[4]pyrrole was confirmed by ^1^H NMR and ^13^C NMR spectra obtained using Bruker Avance III 400 MHz (Billerica, MA, USA) and by FT-IR spectrum obtained using Thermo Nicolet Nexus (Waltham, MA, USA). 

^1^H NMR (400 MHz, CDCl_3_, δ, ppm): 6.97 (s, 1H, NH), 5.88 (d, 2H, PyH, J = 2.7 Hz), 1.49 (s, 6H, CH_3_). ^13^C NMR (100 MHz, CDCl_3_, δ, ppm): 29.10 (CH_3_), 35.19 (C(CH_3_)_2_), 102.83 (ArH), 138.43 (Ar).

For pyrrole units, one strong and sharp peak at 3443 cm^−1^ responds to N-H stretching while a less intensive peak at 3105 cm^−1^ corresponds with =C-H stretching from an aromatic ring. The successive peaks at 2970, 2934 and 2870 cm^−1^ reflect C-H stretching bonds from methyl groups. Stretching bonds from pyrrole units and methyl groups indicate presence of *meso*-octamethylcalix[4]pyrrole. 

The ^1^H NMR, ^13^C NMR and FT-IR spectra are provided in Appendix A.

### 2.3. Preparation of Polymer Inclusion Membranes and Stability Test

The PIMs were prepared according to the procedure previously described [11]. The applied solutions (in dichloromethane as an organic solvent) comprised the following cellulose triacetate, *o*-nitrophenyl octyl ether and *meso*-octamethylcalix[4]pyrrole as ion carrier. A glass ring with a diameter of 5.0 cm connected to a glass plate with a CTA glue-dichloromethane was filled with the predetermined volumes of CTA solution, plasticizer and carrier mixture and left overnight in room temperature to evaporate DCM. Wetting in cold water allowed for the detachment of the resultant membrane from the glass plate. The effective surface area of the CTA membrane was of 4.9 cm^2^, while its average thickness of 24 μm was determined (accuracy of 1.0 μm standard deviation over four readings) by an A2002M type digital ultrameter from Inco-Veritas.

Atomic force microscopy (AFM) (Digital Instruments Veeco Metrology Group, New York, NY, USA) was applied to determine the surface characteristics of polymer membranes, while the analysis of the pore characteristics was performed by the image processing program AFM NanoScope v.720 (Digital Instruments Veeco Metrology Group, New York, NY, USA), which enabled the roughness (Ra) parameter to be calculated as the standard deviation of the *Z* values within the box cursor according to the following formula: (1)$$R_{a}=\frac{1}{N}\overset{N}{\underset{i=1}{\sum }}\left|Z_{i}\right|$$
where *Z_i_* is the current z value, and *N* is the number of points within the box cursors

The principle of operation of AFM consisted in measuring the magnitude of the static deflection of the lever during the sweep of the lever on the sample surface. The topography of the sample surface (change of the coordinate of the axis perpendicular to the sample surface) affects the angle of inclination of the lever and, thus, changes the intensity of illumination of the photodetector sectors. The compensation voltage changes are equivalent to the relative height of the roughness on the sample surface.

Membrane samples were frozen in liquid nitrogen, which resulted in rapid fracturing and in a clean break of fracture image to observe the cross-section. The samples were fixed to metal stubs with conductive glue, with the fractured edge up, and then coated with gold by spraying, and they were examined by a 5 kV scanning electron microscope (SEM) (Quanta 3D FEG, FEI Company, Hillsboro, OR, USA) (Hitachi S4500) to analyze membrane morphology at the magnification of around 200 µm.

To investigate the stability of PIMs in terms of mass loss, which is related to leaching of the carrier and/or plasticizer, they were immersed in 100 cm^3^ of ultrapure water and shaken for 24 h. The mass losses were calculated as a difference between membranes weights before and after this procedure. The masses of membranes were also measured before and after mercury ions transport across PIMs.

The measurement of the contact angle of membranes was carried out with the drop shape method using a Tracker tensiometer (IT Concept, Saint-Ouen-l’Aumône, France). Each membrane was placed on a level table, and then 3 µL of deionized water was dispensed onto the surface of the membrane. Afterward, the angle between the membrane surface and the tangent surface to the droplet at the point of contact with the membrane was examined with the help of a camera directed at the table. The contact angle was read after 300 s for the drop that was provided (the measurement was considered to be stabilized then).

### 2.4. Transport Studies

The experiment in terms of transport was performed at a room temperature (23–25 °C) in a two-compartment permeation cell described in our previous study [11], where the membrane film was closely attached between the source and receiving phases. The source phase comprised an aqueous solution of Hg(II) (50 cm^3^) in hydrochloric acid media, whereas the role of aqueous receiving phase was performed by solutions of sodium chloride, potassium iodide or distilled water (50 cm^3^). Both source and receiving aqueous phases were synchronously stirred at 600 rpm, and aqueous phases were sampled periodically with the use of a sampling port with a syringe for analyzing Hg(II) concentrations. CX-731 Elmetron, a multifunctional pH-meter, was used to control the acidity of both aqueous phases, with a combined pH electrode, ERH-136, Hydromet, Poland. 

The kinetics of the transport process through PIM was described by a first-order reaction with respect to metal ion concentrations [12]:(2)$$\ln\left(\frac{c}{c_{0}}\right)=-kt$$
where *c* is the metal ion concentration (mol/dm^3^) in the source phase at a given time, *c*_0_ is the initial Hg(II) concentration in the source phase (mol/dm^3^), *k* is the rate constant (s^−1^) and *t* is the transport time (s).

The *k* value was calculated from a graph representing the linear relationship of ln(*c*/*c*_0_) versus time (Figure 2), as confirmed by high values (≥0.99) of determination coefficients (r^2^), whereas the permeability coefficient (*P*) was obtained from the following equation:(3)$$P=-\frac{V}{A}k$$
where *V* is the volume of aqueous source phase (m^3^), and *A* is the surface area of membrane (m^2^). The initial flux (*J*_0_) in (µmol/m^2^s) was determined as equal to the following.
(4)$$J_{0}=Pc_{0}$$

The removal efficiency of metal ions from the source phase, described by the recovery factor (*RF*), was calculated as follows.
(5)$$RF=\frac{c_{0}-c}{c_{0}}\cdot 100\%$$

The mercury analyzer AMA 254 (Altec, Dvur Kralove, Czech Republic) was applied to measure the concentration of Hg(II) ions, the other metal ions concentrations were analyzed by flame atomic absorption spectrometry (Solar 939, Unicam, Munich, Germany), whereas chloride and sulphate concentrations were analyzed by ion chromatography (861 Advanced Compact IC, Metrohm, Herisau, Switzerland).

The obtained values correspond to the average of duplicates with the standard deviation within 2%.

## 3. Results and Discussion

### 3.1. Effect of Membrane Composition 

The main factors influencing the efficiency of Hg(II) ions separation during transport through PIMs include membrane composition and the selection of the right receiving phase [26]. The effect of the ion carrier concentration on transport of Hg(II) ions across PIM was examined. Membranes were used with a fixed CTA content (25 mg–19 wt.%) and plasticizer (77 wt.% of *o*-NPOE), while concentrations of the carrier (KP) in the membrane ranged within 1–8 wt.% (0.01–0.30 M) based on the plasticizer’s volume. 

The carrier is of substantial importance to facilitate the transport of Hg(II) ions through the membrane. This was certified with the results of blank experiments (in the absence of carrier), which proved insignificant flux across PIM to contain only the support and plasticizer. Therefore, within the membrane phase, the facilitated transport of mercury was conducted in the presence of the carrier (extractant) [31]. The removal of Hg(II) by means of calix[4]pyrrole is probably due to the obtainment of mercury ions by inclusion complexation inside the cavity forced by the methyl CH_3_ groups. A suggested, the mercury–calix[4]pyrolle complex transported through the membrane is shown in Figure 3.

The transport of Hg(II) increased as the carrier concentration increased (Figure 4) until a concentration of 0.1 M (4 wt.%) was attained, and then the highest flux value of 1.6 × 10^−3^ µmol/m^2^s was obtained. Lower fluxes at higher carrier concentrations probably resulted from the saturation of membrane pores with metal complexes. An increase in the carrier content in the membrane resulted in a decrease in membranes permeability; thus, the value of initial Hg(II) ions flux was obtained probably due to an increase in the curve of the diffusion path of metal complexes caused by an increase in tortuosity factor in the plasticized membrane. The separation of mercury in the PIM was a 3-step process: (i) metal ions were transmitted from the source phase to the boundary layer of the membrane, (ii) the metal-extractant complex was formed at the source solution–PIM interface and (iii) the metal-carrier complex was transferred through PIM [32], and Hg(II) was released at the PIM–receiving solution interface. The carrier reacted with mercury ions to form a complex that was transported across the membrane due to the pH gradient between the source phase and the receiving phase, which was the driving force for metal ion transport by PIM [33]. The highest RF (i.e., 91.8%) was obtained for the membrane with a carrier concentration (KP) of 0.10 M (Figure 4). The lower RF values for carrier concentrations above 0.10 M can be attributed to decreased membrane permeability, which, in turn limited the diffusion of the Hg(II)−carrier complex within the membrane phase. Therefore, the KP concentration of 0.10 M was considered to be optimal. 

In order to study the effect of the plasticizer amount in PIMs on Hg(II) ion transport, the membranes containing a constant carrier concentration of 0.10 M and different contents (0.5–6.0 cm^3^) of *o*-NPOE/1.0 g CTA (63–87 wt.%) were used. In Figure 5, the dependence between RF and the amounts of *o*-NPOE is presented: Along with an increasing amount of plasticizer, the fluxes increased to reach a specific value of the plasticizer in the membrane (4.0 cm^3^ *o*-NPOE/1.0 g CTA—77 wt.%). An increase in PIM rigidity can be defined as an increase in plasticizer amount, which is likely to have reduced the permeability of the membrane and, consequently, the diffusion rate of metal complexes through the membrane; finally, Hg(II) removal performance is reduced as well.

The PIM for which the highest (91.8%) RF factor of Hg(II) was gained was herein after referred to as “optimal” and consisted of 19 wt.% of cellulose triacetate as the support, 4 wt.% of KP and 77 wt.% of *o*-nitrophenyl octyl ether as the plasticizer.

For the membrane with an “optimal” composition, 2-D and 3-D scans were obtained using atomic force microscopy (AFM), which are presented in Figure 6. A vertical profile of the samples is reflected by color intensity: the zones of light point out the highest spikes while the pores (organic inclusion in the CTA support) can be depicted as small well-defined dark regions. Instead, the CTA membrane itself is of non-porous nature and only slightly wrinkled. The middle-sized pores of 0.05 µm were estimated for the “optimal” PIM.

The porosity of the membrane is understood as the proportion of the liquid phase (plasticizer + solvent) on the membrane surface. The pores of the membrane in PIM are the inclusion of the liquid phase (the carrier solution in the plasticizer) into the CTA matrix, creating a transport route of various tortuosity. Thus, the degree of PIM porosity depends on the amount of plasticizer immobilized to the CTA matrix. Typically, such an inclusion has the shape of pores visible on the surface, which is confirmed by SEM and AFM photos for membranes containing *meso*-octamethylcalix[4]pyrrole as ion carrier. 

The SEM images were made in order to observe the possible degradation of the surface and to determine the deposits produced after Hg(II) ion transport across PIM (Table 1). 

A comparison of surface microstructure shows the diversity of matrix materials of the membrane with calix[4]pyrrole derivative before and after the transport of Hg(II) ions across PIM. Crystalline structures of deposits observed in the membrane after the transport process are most probably the result of Hg(II)-calix[4]pyrrole-Cl^−^ adducts or the metal–carrier complex formation. Evident changes in the membrane appearance after transport can be the effect of metal ion accumulation in the membrane phase [34]. The SEM images indicate the formation of deposits, which clog the membrane pores.

Testing the stability of the “optimal” PIM aimed at examining the degree to which its components (the carrier and plasticizer) were leached from the membrane to ultrapure water. The mass loss of membrane was determined at 3% ± 0.2% (n = 2), indicating proper resistance of its components to migration from the base polymer. When pH is below 5 and above 7, the plasticizer (*o*-nitrophenyl octyl ether), as a representative of the phthalate esters group, is subject to hydrolysis, the extent of which increases with a pH increase or decrease and with longer contact times [35]. 

In this study, membrane mass loss at the acidic source phase after 8 h of Hg(II) transport across the PIM was of 5% ± 0.3% (n = 2), showing the proper binding of the plasticizer to the polymer. Otherwise, when meaningful leaching may occur, due to toxicity of the phthalates [36], using a plasticizer at the pilot or industrial scales is not required.

The hydrophobicity of membranes, defined as the wetting, can affect the parameters of metal ions transport. Wetting provides an information about hydrophobic properties of the membrane surface, which is related to both matrix and immobilized organic phase containing a carrier [37]. The measurements of wetting contact angles before and after long-term transport across PIMs were performed in order to determine changes in membranes surface hydrophobicity. Table 2 presents the appearance of a water drop on the membrane before and after Hg(II) ions transport through the PIM and the changes in measured values of wetting contact angles.

The wetting contact angle for the PIM containing calix[4]pyrrole derivative was of 25.26°. This value proves a hydrophilic character of the CTA membrane as no wetting contact angle above 90° was observed [38]. The surface properties of the PIM depend on the carrier effect on the formation of inclusions (pores) of liquid organic phase in a solid CTA matrix [37]. During mercury transport across the PIM, an increase in the wetting contact angle from 25.26° to 48.26° was noted, which suggests the occurrence of metal complexes deposits on the membrane’s surface or elution of the plasticizer.

### 3.2. Modification of the Source Phase Acidity

The investigation also included examining the influence of the concentration of hydrochloric acid in the source phase within the range of 0.01 ÷ 1.0 M on Hg(II) ions transported through PIM. Along with the increasing acidity in the source phase, the efficacy of mercury separation was increasing until reaching a maximum value of 91.8% at the HCl concentration of 0.1 M (Figure 7). The pH gradient induced diffusion of the Hg(II)−carrier complex through the membrane, promoting its release from the membrane [39]. In turn, an increase in chloride content tends to reduce RF of Hg(II) due to the occurrence of anionic mercury chloride species in the source solution. An increase in chloride content in the source phase is responsible for the reduced mobility of the extracted species or the release of the plasticizer to the membrane/aqueous surface [40].

### 3.3. Membrane Reusability 

Among various properties, the greatest advantage of PIMs is the possibility of their reuse, which ensures its industrial usefulness [10]. During the transport of a six-cycle duration with Hg(II) ions across the membrane, both aqueous phases were renewed each time while the “optimal” PIM remained the same as in the first pass. The results show the reproducibility of PIM performance (Figure 8): RF values were above 91% in the first four transport cycles (each cycle of 8 h), and then they were slightly decreased (RF of ca. 89%). Therefore, the “optimal” PIM appears to be effective for repeatable use in the case of mercury removal from aqueous waste streams. 

### 3.4. Effect of the Type of Agents in the Receiving Phase

What significantly affects the performance of mercury ions transport across PIMs can include the nature and composition of the receiving phase. With different strip agents, the percentage of transported Hg(II) ions under similar experimental conditions is listed in Table 3.

While applying 0.1 M NaCl as the receiving phase, the most effective Hg(II) ion transport through PIMs was reached compared to other strip agents. The sum of percentages of mercury ions in the source and receiving phases is not equal to 100; there can be some Hg(II) ions staying in the membrane, particularly when the stripping agent does not have enough tendency to release Hg ions completely from the Hg(II)-calix[4]pyrrole-Cl^−^ adducts (included in the membrane phase) into the receiving phase.

### 3.5. Removal of Hg(II) from Zinc Smelting Wastewater

The applicability of the “optimal” PIM was also investigated for the separation of mercury ions from zinc smelter wastewater. The sample was characterized as the acidic solution with a very high total dissolved solids concentration due to heavy metal ions (Cu(II), Pb(II), Zn(II), Cd(II) and Hg(II)), chloride and sulphate ions [41]. Hg(II) separation from the wastewater performed in the “optimal” PIM with 0.1 M sodium chloride as the receiving phase resulted in a removal efficacy of 85.7% after 8 h of continuous operating time. The initial Hg(II) concentration declined from 0.14 to 0.02 mg/dm^3^, i.e., below the permissible value in treated wastewater in accordance with Polish legislation. Moreover, the RF values found for other constituents of wastewater were below 7%, which demonstrate the selectivity of the system for mercury ions (Figure 9).

## 4. Conclusions

The efficiency of mercury ion separation with the use of PIMs depends mainly on the acidity of the source phase, membrane’s elements (concentration of the carrier and plasticizer in the PIM) and selection of the appropriate receiving phase. The optimization of PIMs composition and selection of appropriate process parameters allow achieving their high efficiency and stability for the selective removal of mercury ions from industrial wastewater. The arrangement of the “optimal” PIM followed from our study was 19 wt.% of cellulose triacetate as the support, 4 wt.% of KP as the ion carrier and 77 wt.% of *o*-nitrophenyl octyl ether as the plasticizer. The highest Hg(II) ions separation efficacy of 91.8% was obtained at the concentration of HCl in the source phase of 0.1 M and when 0.1 M sodium chloride was used as the receiving agent. Moreover, the membrane was reusable and showed a good performance (85.7% removal) in selectively removing mercury ions from the zinc smelting wastewater.

The separation of heavy metals in membrane processes with the use of PIMs can significantly improve the most technologically important operations of recovering metals from wastewater, thus solving the technical and economic problems of removing toxic pollutants from aqueous waste streams for the benefit of the natural environment.

## Figures and Tables

**Figure 1 membranes-12-00492-f001:**
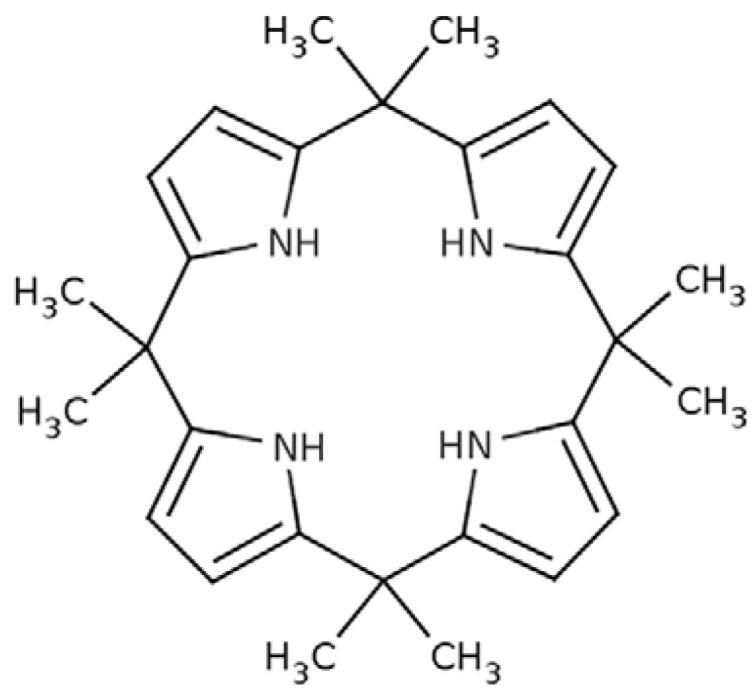
Structure of *meso*-octamethylcalix[4]pyrrole (KP).

**Figure 2 membranes-12-00492-f002:**
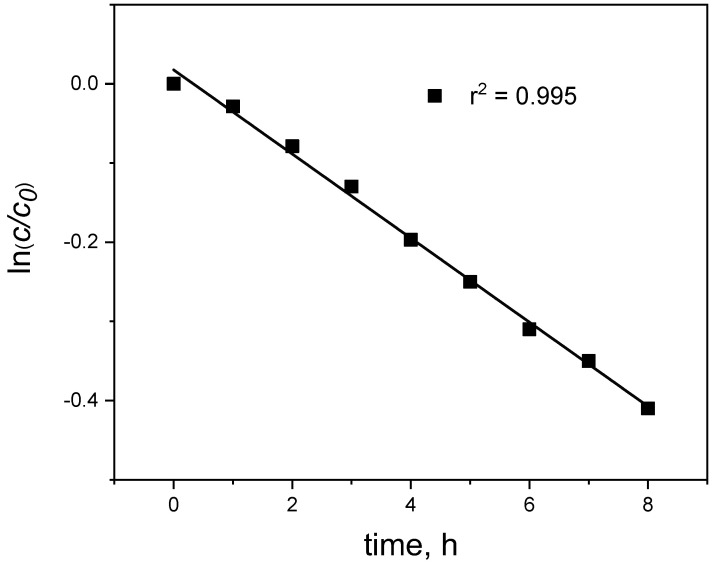
The relationship of ln(*c/c*_0_) vs. time for mercury transport across PIM. Source phase: 1.0 × 10^−6^ M Hg(II), 0.1 M HCl; membrane: 19 wt.% of CTA, 77 wt.% of *o*-NPOE; receiving phase: 0.1 M NaCl.

**Figure 3 membranes-12-00492-f003:**
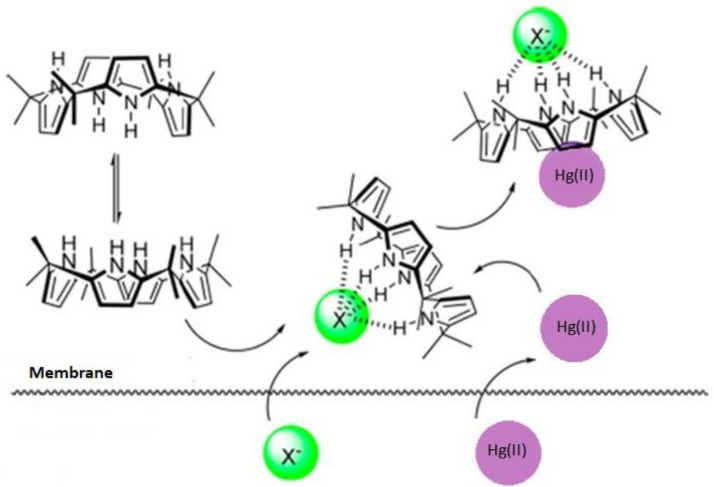
A suggested mercury–calix[4]pyrrole (KP) complex transport mechanism through PIM membrane (X^−^ = Cl^−^).

**Figure 4 membranes-12-00492-f004:**
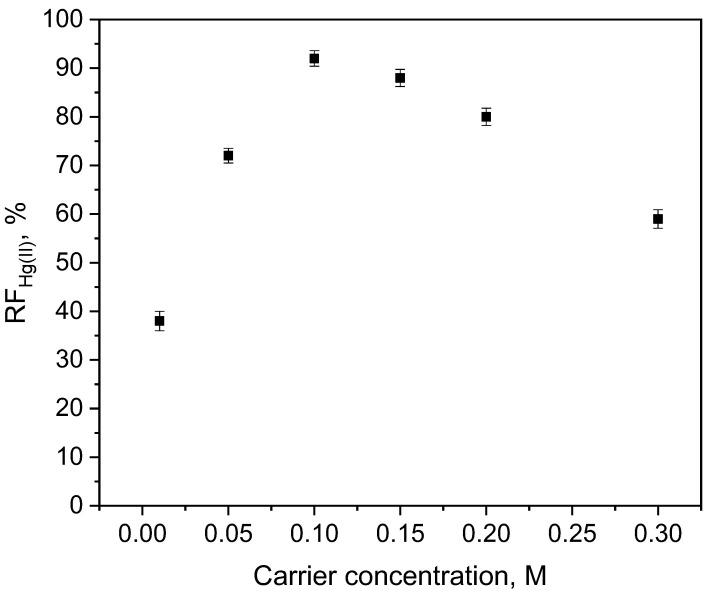
The recovery factor (RF) of Hg(II) ions vs. the carrier concentration. Source phase: 1.0 × 10^−6^ M Hg(II), 0.1 M HCl; membrane: 19 wt.% of CTA, 77 wt.% of *o*-NPOE; receiving phase: 0.1 M NaCl.

**Figure 5 membranes-12-00492-f005:**
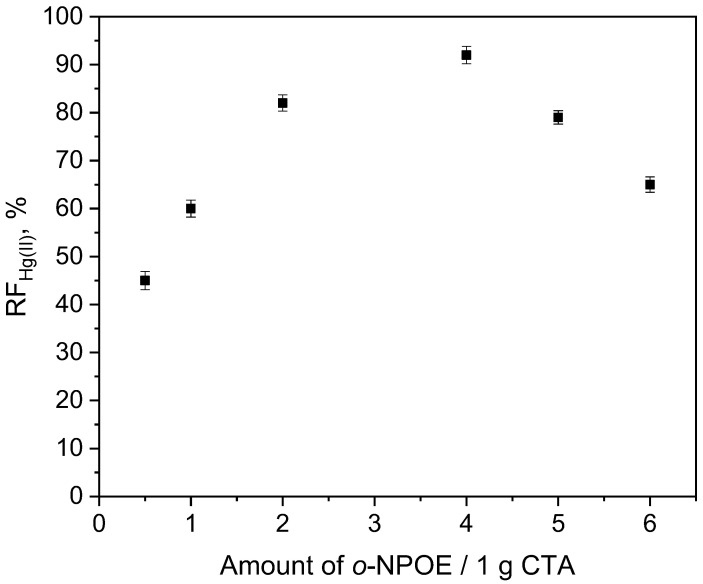
The effect of *o*-NPOE plasticizer on the recovery factor of Hg(II) transport through PIM containing KP. Source phase: 1.0 × 10^−6^ M Hg(II), 0.1 M HCl; membrane: 19 wt.% of CTA, 4 wt.% of KP; receiving phase: 0.1 M NaCl.

**Figure 6 membranes-12-00492-f006:**
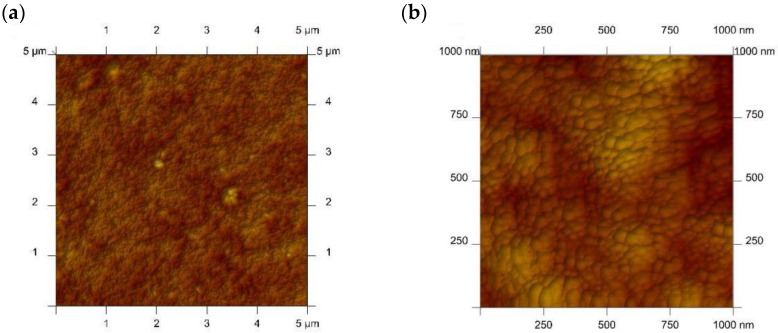
The 2-D (**a**,**b**) and 3-D (**c**,**d**) images of atomic force microscopy (AFM) for the CTA membrane (**left**) and the “optimal” polymer inclusion membrane (PIM) (**right**).

**Figure 7 membranes-12-00492-f007:**
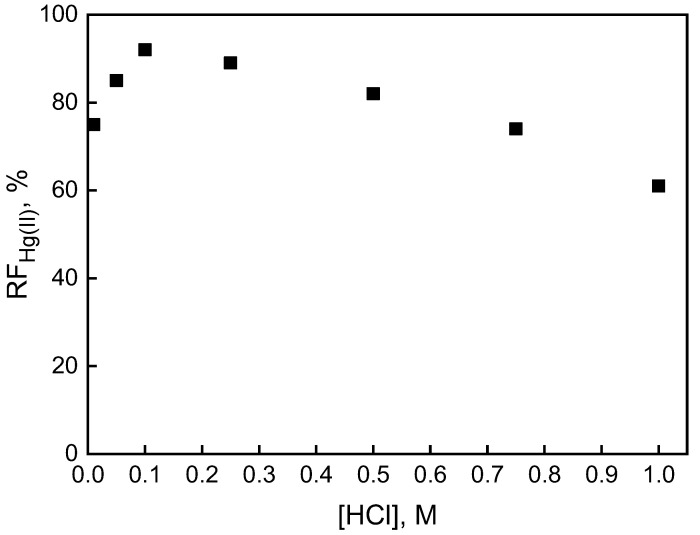
The recovery factor (RF) of Hg(II) ions vs. HCl concentration in the source phase. Source phase: 1.0 × 10^−6^ M Hg(II); membrane: 19 wt.% of CTA, 4 wt.% of KP, 77 wt.% of *o*-NPOE; receiving phase: 0.1 M NaCl.

**Figure 8 membranes-12-00492-f008:**
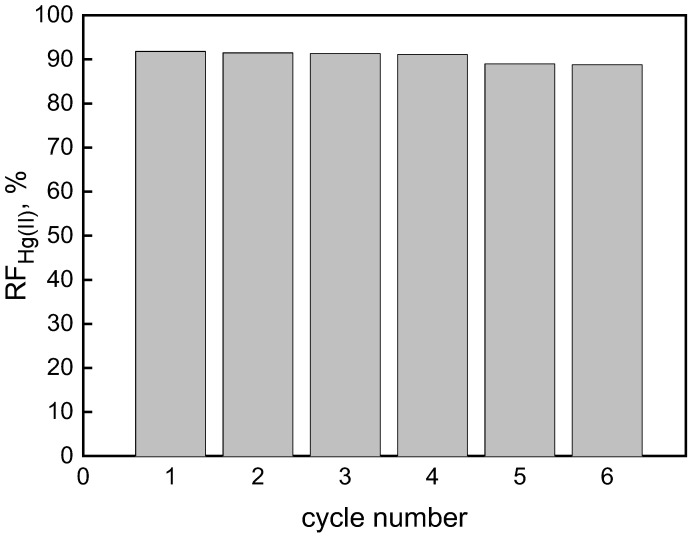
The recovery factor (RF) of Hg(II) vs. the cycle number. Source phase: 1.0 × 10^−6^ M Hg(II), 0.1 M HCl membrane: 19 wt.% of CTA, 4 wt.% of KP, 77 wt.% of *o*-NPOE; receiving phase: 0.1 M NaCl.

**Figure 9 membranes-12-00492-f009:**
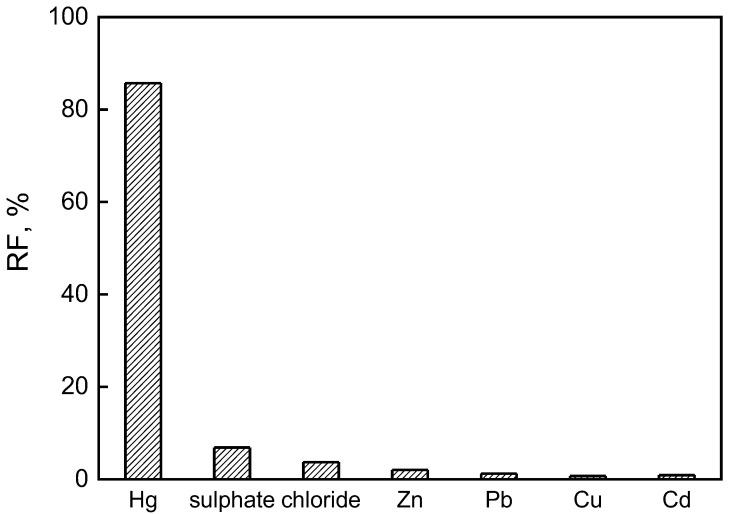
The RF values obtained in the competitive PIM transport of mercury from zinc smelting wastewater sample. Membrane: 19 wt.% of CTA, 4 wt.% of KP, 77 wt.% of *o*-NPOE; receiving phase: 0.1 M NaCl.

**Table 1 membranes-12-00492-t001:** SEM images before and after the transport of Hg(II) ions across PIM containing KP. Source phase: 1.0 × 10^−6^ M Hg(II); membrane: 19 wt.% of CTA, 4 wt.% of KP, 77 wt.% of *o*-NPOE; receiving phase: 0.1 M NaCl.

Before Transport of Hg(II) Ions	After Transport of Hg(II) Ions
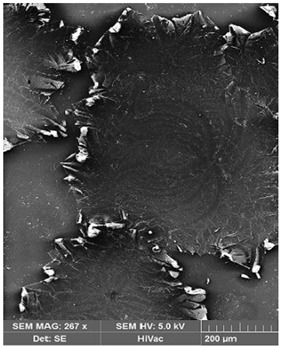	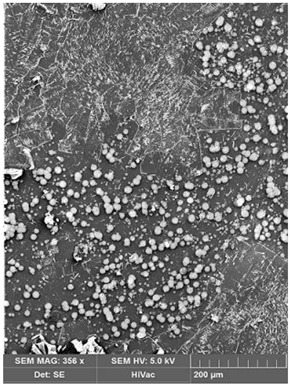

**Table 2 membranes-12-00492-t002:** The appearance of a water drop on the membrane and the contact angles of the polymer matrices before and after the transport of Hg(II) ions across PIM. Source phase: 1.0 × 10^−6^ M Hg(II); membrane: 19 wt.% of CTA, 4 wt.% of KP, 77 wt.% of *o*-NPOE; receiving phase: 0.1 M NaCl.

Before Transport of Hg(II) Ions across the PIM	After Transport of Hg(II) Ions across the PIM
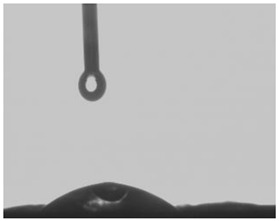 **25.26°**	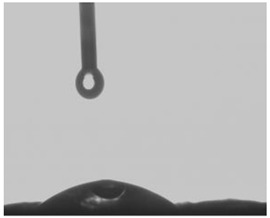 **48.26°**

**Table 3 membranes-12-00492-t003:** Effect of the nature of the receiving phase on the transport of Hg(II) ions across PIMs. Source phase: 1.0 × 10^−6^ M HgCl_2_; membrane: 4.0 cm^3^ o-NPOE/1.0 g CTA; 0.1 M carrier (KP).

Receiving Agent	Percentage of Hg(II) Transported into the Receiving Phase	Percentage of Hg(II) Remaining in the Receiving Phase
0.1 M NaCl	92	4
0.1 M KI distilled water	67 54	14 25

## Data Availability

Not applicable.

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
