# Peer review of "Separation of Mercury(II) from Industrial Wastewater through Polymer Inclusion Membranes with Calix[4]pyrrole Derivative"

_membranes, 2022, doi:10.3390/membranes12050492_

Round 1
Reviewer 1 Report
The article discusses development of new PIM membranes impregnated with cyclic chelating ion carriers for removal of Hg(II) ions from waste water streams. The authors have tested different parameters such as the source phase acidity, amount of carrier and the plasticizer in the membrane and how they effect the Hg removal efficiency and eventually identified the optimal membrane composition that removes >90% of Hg(II) ions from the source. The practical applications of such membranes will have high impact and this work could be significant in addressing some key challenges in wastewater remediation, not only in the sectors that authors mention in the introduction section but also other areas such as Flue Gas desulphurization streams which contain Hg and other heavy metal traces above the discharge limits. However, there are some gaps in the manuscript related to experimental design as well as the interpretation of the data, that need to be addressed before the manuscript is recommended to be accepted for publication.
- Line 27, please change to "..sixth most toxic.."
- Line 27, there is a random "c" in the text. please delete.
- Figure 2, please highlight the relevant IR peaks along in the figure for easy understanding.
- Lines 110-113, please specify the measurement mode, tip used and the scanning rate. Also, please describe the pore characterization method in further detail.
- How was the pore size determined? and
- please present a 2D image used for the image analysis.
- Line 132,
More details on the permeation cell tests need to be included.
-
Were the tests carried out with recycling of the source phase or was it a single pass where a given volume of feed water is passed through the cell once while monitoring the change in concentration of Hg ions?
-
What was the flow rate of the source phase?
-
- Lines 211-212, The actual pH of the source and the receiving phases are not mentioned anywhere. What was the actual pH gradient in the experiments? Did the authors attempt varying the pH gradient to see the change in Hg ion flux?
- Line 215, How is membrane viscosity quantified? Is there a similar observation in the literature or is this a hypothesis at this stage? Authors should specify that this is a hypothesis. How does the bulk viscosity effect other physio chemical properties of the membrane, such as the mechanical properties? If experimental validation is not possible, please state clearly that this is a hypothesis and also add any relevant references, if any.
- Figures 5 and 6, the data points need error bars.
- Lines 245-246, The use of AFM images for pore size characterization is not very quantitative because AFM height images (in both contact and tapping modes) can vary depending upon the force used (contact mode) or scanning rate (tapping/contact mode) used. If possible, the authors should rely on more accurate methods such as porometry or BET measurements to determine the pore size distribution.
- Line 293 - 299, It is not clear how the hydophobicity (or hydrophilicity) specifically effects the ion transport properties. There is no strong correlation between the contact angle data presented and the explanation given about "increase in contact angle confirms the occurrence of metal complexes.." Does it mean that more hydrophilic surface enables the ion permeation better? please clarify with any available literature precedence or please delete the section on the contact angles.
- Conclusions, does the removal efficiency also depend on the Hg ion concentration in the source phase? Please add a statement mentioning the effect of Hg ion concentration on removal efficiency.
Reviewer 2 Report
In this manuscript, the authors evaluate the ability of polymer inclusion membranes (PIMs) with a calix[4]pyrrole derivative as carrier to remove mercury (Hg(II)) ions from wastewater. The parameters influencing the removal efficiency such as carrier concentration, plasticizer amount in the membrane, source phase acidity and type of receiving phase are explored. Also, the reusability of membrane is investigated. Some valuable results are obtained. This manuscript is recommended with minor revision.
Detailed comments are as follows:
1. Page 5, line 181: The presentation to the role of plasticizer and base polymer is better moved to the Introduction. Furthermore, why o-NPOE was selected as plasticizer and CTA as base polymer to prepare PIMs? Please include these information in the Introduction.
2. Page 6, lines 201-203: The authors consider that the value of initial Hg(II) ions flux significantly decreased with an increase in membrane thickness as a result of higher carrier content. What thicknesses were the manufactured membranes? It is best to show the thicknesses of the PIMs in the manuscript.
3. Page 6, Lines 203-205: This sentence is unclear.
4. Page 7, line 229: What thicknesses were the manufactured membranes with different amount of plasticizer?
5. Page 9, line 262: Discuss more the comparison between SEM images before and after the transport of Hg(II) (Table 1).
6. Page 9, line 279: Please add suitable references regarding the statement “The hydrophobicity of membranes, defined as the wetting, can affect the parameters of metal ions transport.”
7. Page 10, line 297: Can the authors suggest a reason why the value of wetting contact angle after the mercury transport process was increased?
8. Page 12, line 337: Can the author give a reason why the mercury transport is the most effective when using 0.1 M NaCl as the receiving phase?
